# Deoxynivalenol (DON) Accumulation and Nutrient Recovery in Black Soldier Fly Larvae (*Hermetia illucens*) Fed Wheat Infected with *Fusarium* spp.

**Zehra Gulsunoglu** [1,2] 🆔, **Smitha Aravind** [2], **Yuchen Bai** [2], **Lipu Wang** [3] 🆔, **H. Randy Kutcher** [3] 🆔 **and Takuji Tanaka** [2,*] 🆔

1   Faculty of Chemical and Metallurgical, Food Engineering Department, Istanbul Technical University, Istanbul 34469, Turkey
2   Department of Food and Bioproduct Sciences, University of Saskatchewan, 51 Campus Dr. Saskatoon, SK S7N5A8, Canada
3   Department of Plant Sciences, University of Saskatchewan, 51 Campus Dr. Saskatoon, SK S7N5A8, Canada
*   Correspondence: takuji.tanaka@usask.ca; Tel.: +1-306-966-1697; Fax: +1-306-966-8898

**Abstract:** Fusarium head blight (FHB) is one of the most significant causes of economic loss in cereal crops, resulting in a loss of $50–300 million for Canadian agriculture. The infected grain (containing *Fusarium*-damaged kernels (FDKs)) is often both lower in quality and kernel weight, and it may be unsuitable for human and animal consumption due to mycotoxin presence. However, it still contains a considerable amount of nutrients. A method to recover the nutrients without the mycotoxins should be beneficial for the agricultural economy. In this study, our objective was to examine recovery methods of the nutrients in relation to mycotoxin accumulation in the insect. The FDKs were fermented with *Aspergillus oryzae* and/or *Lactobacillus plantarum* (solid-state fermentation (SSF)). The SSF kernels were then provided to 50 young, black soldier fly larvae (BSFL) for 12 days. Weight gain, chemical composition, and mycotoxin bioaccumulation of BSFL and spent feed were evaluated. After 12 days of insect culture, the BSFL grew 5–6 times their initial weight. While the overall weights did not significantly vary, the proteins and lipids accumulated more in SSF FDK-fed insects. During the active growth period, the larval biomass contained deoxynivalenol (DON), a mycotoxin, at detectable levels; however, by day 12, when the larvae were in the pre-pupal stage, the amount of DON in the insect biomass was nearly negligible, i.e., BSFL did not accumulate DON. Thus, we conclude that the combination of BSFL and SSF can be employed to recover DON-free nutrients from FHB-infected grain to recover value from unmarketable grain.

**Keywords:** insect culture; solid-state fermentation; mycotoxins; value-added processing; Fusarium head blight

## 1. Introduction

Fusarium head blight (FHB) is a fungal disease caused by several *Fusarium* spp. Wheat, barley, oats, corn, and other cereal grains can be affected by FHB, resulting in small lightweight kernels and, thus, loss of yield. *Fusarium* spp. produce various amounts and types of trichothecene mycotoxins, which are highly toxic to humans and livestock [1]. A major mycotoxin produced by *Fusarium* spp. is deoxynivalenol (DON). Toxin production occurs during disease development in the field under favorable weather conditions. Contamination of food and feedstuff with DON causes short- and long-term adverse effects on human health and livestock productivity [2]. In order to limit the mycotoxins in food and feed, regulations specify maximum allowable concentrations, which is 1 mg/kg sample in many countries. According to the regulations, products are monitored, and when mycotoxin

concentrations exceed the maximum allowable limits, products are separated from the food chain [3]. The economic loss from FHB accounts for many millions of dollars in Canada alone.

Detoxification methods are expensive, labor-intensive, inefficient, and time-consuming, and there is inadequate capacity for industrial applications. An effective way to prevent FHB in the field is to treat the flowering wheat plants with fungicides, and to develop resistant cultivars to minimize the infection of *Fusarium* spp. Fungicide, however, has limited effects on the infection, and, every year, a large number of grains are damaged by FHB. Considering that possible approaches to prevent the contamination of grain with mycotoxins are limited before harvest, alternate approaches should be considered to utilize inedible FHB-damaged kernels (FDK). In this study, we aimed to investigate if black soldier fly larvae (BSFL) can grow on FDK without any toxin accumulation in larval body.

World population is increasing, and it is predicted to reach 9.6 billion by 2050, i.e., a 2.3 billion increase in the next 30 years. Food production relies on agriculture, but the current practice in agriculture may not be sufficient to supply enough food for this population increase, without damaging Mother Earth or introducing super-high-yield crops that do not result in environmental damage. Utilization of inedible agriculture products can bypass the above concerns and can yield additional edible products from current practice agricultural production [4].

There is a considerable interest in the use of insects to recover inedible organic matter because insects can convert carbohydrates into proteins and lipids using organic wastes [5]. *Hermetia illucens* (black soldier fly) is one of the most important species, along with other insect species like *Tenebrio molitor* (yellow mealworm), *Drosophila melanogaster* (common fruit fly), *Amyelois transitella* (orange worm), *Helicoverpa zea* (corn earworm), and *Trichoplusia ni* (cabbage looper) [3]. While the regulations vary among countries and areas, the nutrient values and ease of utilization of insect nutrients draw huge interests for providing an alternative food source [6]. BSFL are considered a possibly proteinaceous animal feed or human food source because of the high accumulation of fat (29%) and protein (42%) in their body, and they do not transmit pathogenic microbes to humans and animals [7]. BSFL have high feed conversion ratios and an ability to convert various organic wastes into body mass [8,9].

Solid-state fermentation (SSF) shows great possibilities in the development of high-value products. Fungal and bacterial strains can be used in SSF, utilizing their abilities of enzyme production such as cellulase, pectinase, and xylanases. In this study, we used *Aspergillus oryzae* and *Lactobacillus plantarum* as microbial strains based on data obtained from our previous study [4]. The main objective of SSF was increasing the bioavailability of nutrients in FDK in favor of recovering them as BSFL biomass, and changing the nutrient profiles in favor of improved/efficient nutrient recovery in BSFL. Through these microbial modifications, efficiency of nutrient recovery from damaged crops should be enhanced during BSFL digestion.

The BSFL can be utilized to reduce pollution and convert low-value organic resources into a high-quality feed protein. They do not harbor diseases, and their production does not need any special equipment of facilities. BSFL are an extremely resistant species capable of dealing with demanding environmental conditions, such as drought, food shortage, or oxygen deficiency. The BSFL is already used in the waste management of some substrates such as manure, rice straw, food waste, kitchen waste, distillers' grains, rotting plant tissues, fecal sludge, animal offal, and animal manure [8]. We showed that BSFL can convert agricultural wastes into biomass with up to 95% recovery of organic matter using SSF [4,10]. Our previous research [4,10] suggests that FDK could be fed to BSFL to recover nutrients at a high rate with SSF treatment of feedstock. It is, however, unclear if the mycotoxins would be accumulated when BSFL are fed with FDK. We hypothesize that the BSFL are not affected by the mycotoxin content of wheat grain infected with *Fusarium* spp., they will not accumulate mycotoxins in their bodies, and a majority of the nutrients in the FDK will be recovered using SSF treatments. Efficient recovery of nutrients from damaged crops can be used to create a high-value product from low-value grain. In this study, the performance of BSFL in converting SSF-treated FDK into insect biomass and the accumulation of DON were investigated.

## 2. Material and Methods

### 2.1. Materials

BSFL were purchased from Worm Lady (McGregor, ON, Canada). All chemicals used in this study were commercially available ACS grade and were purchased from Fisher Scientific (Ottawa, ON) and VWR International (Edmonton, AB). *Aspergillus oryzae* NRRL 32657 (*Ao*) and *Lactobacillus plantarum* NRRL B4496 (*Lp*) were obtained from the ARS Culture Collection (USDA, Peoria, IL, USA).

### 2.2. Solid-State Fermentation

The initial DON concentration of FDK was 0.63 ± 0.20 µg/g dry matter (dm). The FDKs were soaked in water for 18 h at 21 °C to obtain softer kernel for fermentation. The soaked kernels were shred using a household coffee mill (Cuisinart DBM-8, Woodbridge, ON, Canada). Seed cultures of *Ao* and *Lp* were prepared by inoculating Potato Dextrose (PD) and De Man, Rogosa, and Sharpe (MRS) broths. The seed culture of *Ao* was prepared by inoculating 250 mL of PD broth in an Erlenmeyer flask with two loopfuls of spores, followed by 72-h incubation at 30 °C on a rotary shaker (150 rpm). The seed culture of *Lp* was prepared by inoculating 1 mL of fully grown preculture in 250 mL of MRS broth in an Erlenmeyer flask and incubating for 24 h at 37 °C on a rotary shaker (150 rpm). After incubation, fungal and bacterial seed cultures were collected by centrifugation at 6000 rpm for 10 min at 10 °C (Sorvall, RC28S, Manasquan, NJ, US). The biomass obtained was re-suspended in 1/10 of the original volume of sterile distilled water.

Approximately 58 g (34 g in dry weight) of shredded kernels were weighed into each glass jar and, from the seed cultures, 1.5 mL of *Lp*, 2 mL of *Ao*, and 3.5 mL of a combination of these two strains (*Lp* + *Ao*) were inoculated into the crushed kernels. The moisture content was adjusted to 55% (*w/w*) with sterile water. The control sample was prepared without initial seed culture inoculation. The samples were fermented at 30 °C for four days, and the moisture content was kept constant by adding sterile water and mixing once per day under aseptic conditions.

### 2.3. BSFL Digestion

Fifty BSFL (second instar) were introduced to each SSF FDK sample in the glass jars that had perforated lids to allow moisture and gas transfer during BSFL digestion. Twelve jars were prepared under the same conditions for each SSF FDK sample. The jars were kept at 30 °C for 12 days. Water was added by weighing the sample jars and mixing each day under aseptic conditions.

### 2.4. BSFL Separation after Digestion

After interval days (0, 4, 8, and 12 days), the larvae were separated from the residual feed using forceps. The larvae were rinsed with water to remove residual substrate from their surface and dried on paper towel; then, their wet weight was determined. Then, the larvae were frozen at −20 °C for further analysis. The rinsed-off feed residues were placed back in the feed bed to avoid errors in spent feed analyses.

### 2.5. Larval Weight Gain Determination and Survival Rate of Larvae

For survival analysis, the number of larvae was counted at 0, 4, 8, and 12 days (initially, exactly 50 larvae). To monitor larval growth, the dry weight of each sample was determined. Larval volume was calculated by multiplying the length, width, and thickness of 10 individual larvae before and after digestion.

### 2.6. Proximate Analysis of BSFL and Spent Feed

Proximate analysis was performed on the BSFL and spent feed before and after larval digestion of the fermented FDK. The dry weight of the samples was measured after drying at 105 °C for 24 h to a

constant weight according to AOAC Method #930.15 [11]. Dried samples were used for ash, crude protein, and crude fat analyses. The ash content of the larval biomass and spent feed was determined by the gravimetric method as described in AOAC Method #942.05 [11]. Samples were carbonized using a hot plate in the fume hood and, after carbonization, the crucibles were incinerated in a muffle furnace at 550 °C overnight. Crude protein content was measured by the micro-Kjeldahl method as described in AOAC Method #960.52 [11] with slight modification. Conversion factors of 6.25 and 5.70 were used to calculate total protein of the larval biomass and spent feed, respectively. The crude fat analysis was determined according to the Goldfisch method as described in AACC method #30-20.01 [12]. Samples were weighed onto Whatman filter paper (No.1) at 0.5 g for larval biomass and 1 g for spent feed and placed in the Goldfisch apparatus (Labconco Corporation, Kansas City, MO, US). Petroleum ether was used as the extraction solvent, and the extraction process lasted 6 h. Crude fat was determined as the weight of fats in the extract after removal of the solvent. The carbohydrate contents of BSFL and spent feed were determined by subtracting the lipid, protein, and ash contents from the total weight.

*2.7. Mycotoxin Analysis*

Larval biomass and spent feed were finely ground and extracted for mycotoxin analysis to determine the concentration of DON accumulated. Larval samples and spent feed were extracted according to the method developed by Dr. L. Wang, at the Cereal and Flax Pathology program at the University of Saskatchewan (personal communication). Finely ground wheat grain (2 g) and a larval sample (0.1 g) were mixed with acetonitrile/water (84:16, *v/v*) with a ratio of 1:4 *w/v* and extracted on a rotary shaker for 2 h at room temperature (250 rpm). The extract was diluted 1:10 with 5 mM ammonium acetate and syringe-filtered. Thirty microliters of this filtrate was injected into the LC–MS/MS.

The LC–MS/MS conditions were developed on a high-performance liquid chromatography system (Agilent 1260 Infinity Quaternary; Agilent Technologies, Mississauga, ON, CA) coupled to an AB Sciex 4000 hybrid triple quadrupole linear ion trap (4000 QTrap) mass spectrometer (Concord, ON, CA) equipped with a Turboionspray$^{TM}$ interface. Applied Biosystems/MDS Sciex Analyst software (Version 1.6.2, AB Sciex, Foster City, CA, US) was used for system control and quantification. The mobile phase consisted of a mixture of solvent A (5 mM ammonium acetate in water) and solvent B (5 mM ammonium acetate in methanol). Samples were stored in the auto sampler at 4 °C and a 30-µL injection volume with a 3-s flush port wash (to minimize carryover) was used to introduce the sample into the column.

Chromatographic separation was obtained at a flow rate of 300 µL/min through an Agilent ZORBAX Eclipse XDB C18 column (4.6 × 100 mm, 1.8 µm) equipped with an Eclipse XDB C18 (4.6 mm, 1.8 µm) guard column maintained at 30 °C in a column heater. Multiple reaction monitoring (MRM) with electrospray was used to monitor DON with the transitions of *m/z* 355.0 to *m/z* 296.1 and *m/z* 355.0 to *m/z* 264.9 as quantifier and qualifier ions, respectively.

The DON standard (purity > 99%) was supplied by Romer Labs Inc. (Tulln, Austria), and the stock solution was prepared at a concentration of 1 µg/mL in acetonitrile and kept at −20 °C. Working stock solutions were made by diluting in 5 mM ammonium acetate to the level of 10-fold final working concentration for each standard and quality control (QC) point. Standard and QC samples were prepared by adding 100 µL of each working stock to 900 µL of blank sample and mixing gently. A standard curve of seven points was constructed by determining the best fit of peak area versus the analyte concentration and running a weighed $1/x$ linear regression analysis.

*2.8. Statistical Analysis*

Each treatment was analyzed in triplicate, and results are presented as an average with standard deviation. Statistical significance of the results was analyzed by one-way analysis of variance (ANOVA) using MINITAB (MINITAB 18, Minitab Inc., Coventry, UK). Treatment means were declared significantly different from each other using Tukey's test at $p < 0.05$.

## 3. Results and Discussion

### 3.1. Growth Rate of BSFL

The amount of feed left after periods of BSFL rearing is given in Figure 1. The BSFL consumed about 65, 45, 61, and 72 mg FDK/larva/day during the first four days for the control, *Lp*, *Ao*, and *Lp + Ao* groups, respectively. In the next four days, the BSFL consumed 67, 82, 76, and 84 mg FDK/larva/day for the control, *Lp*, *Ao*, and *Lp + Ao* groups, respectively. After eight days of consumption of FDK, BSFL growth rates decreased for all treatments. The BSFL consumed 30, 42, 21, and 10 mg FDK/larva/day during the last four days for the control, *Lp*, *Ao*, and *Lp + Ao* groups, respectively.

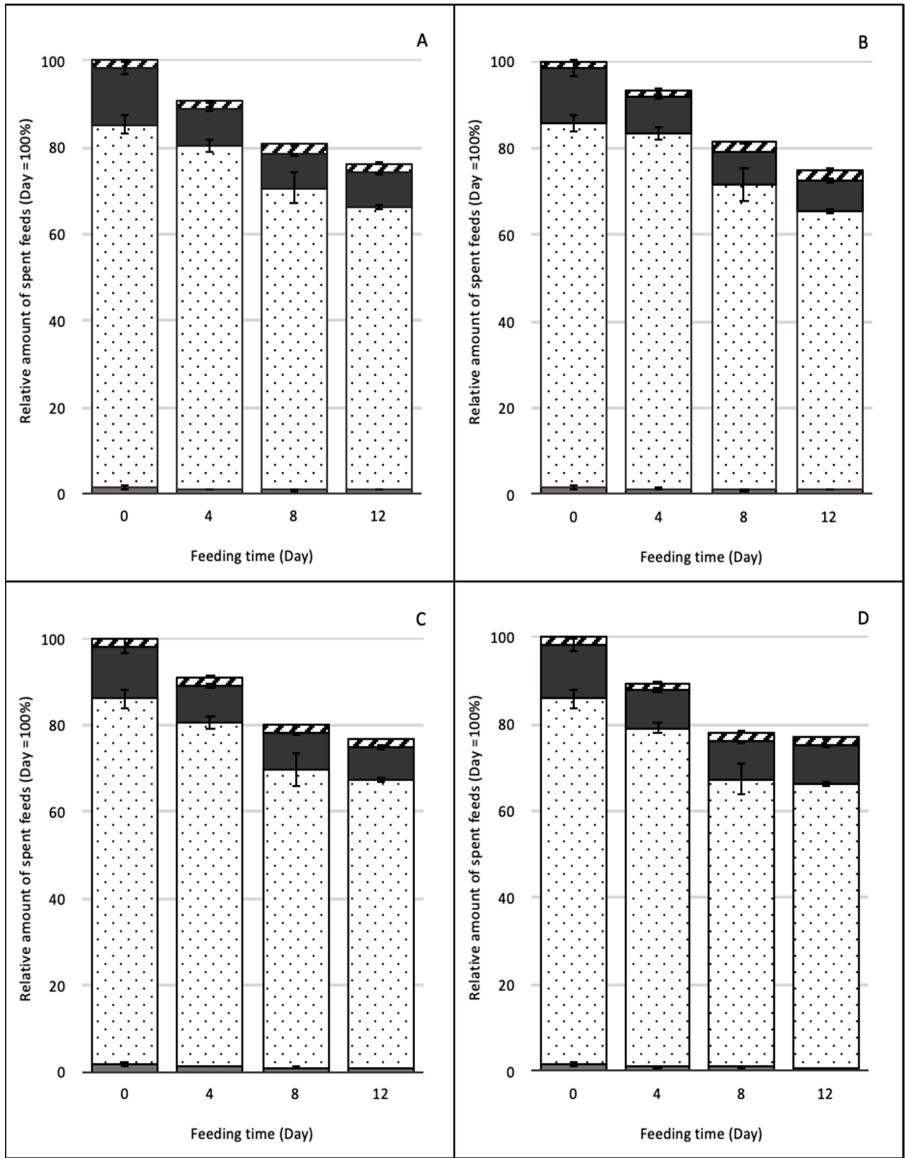

**Figure 1.** The relative amount of spent feed during black soldier fly larvae (BSFL) digestion. Bars indicate differences in nutrient composition of spent feed after BSFL digestion based on initial day. During feeding time, the residual amount was calculated by taking into account the weight of Fusarium-damaged kernels (FDKs) consumed by BSFL. Each bar was divided according to the ratio of nutrients: hatched area, ashes; black area, proteins; dotted area, carbohydrates; gray area, lipids. Each panel represents the results from (**A**) control without any inoculum (i.e., unfermented FDK), (**B**) fermented FDK with *Lactobacillus plantarum*, (**C**) fermented FDK with *Aspergillus oryzae*, and (**D**) fermented FDK with both *L. plantarum* and *A. oryzae*.

Feed consumption was relative to the growth rate of larvae. The weights and volumes of BSFL from each batch fed FDK fermented with *Lp*, *Ao*, and *Lp + Ao* were determined for each feeding time (Figure 2). Increases in weight were observed until day eight for all treatments. From the eighth to the twelveth day, BSFL weight did not differ for any of the treatments. Initial weight of BSFL (day zero) was $8.5 \pm 0.4$ mg on a dry weight basis (dwb) per larva. Larvae gained significantly more weight when fed with FDK fermented with *Lp* ($58.4 \pm 6.3$ mg/larva). Kuttiyatveetil et al. [4] assessed SSF borage meal and flaxseed meal as the feedstuffs and reported that the highest BSFL biomass was 87.4 mg/larva. The volume of BSFL increased from $66.8 \pm 21.8$ mm$^3$ to $352.7 \pm 87.9$ mm$^3$ when fed FDK fermented with *Ao* at day eight, and there were no significant differences among treatments. Larval survival rates did not differ among treatments. At the end of the feeding time, 90–93% of initial larvae survived.

It is speculated that the final four days were the pupation period. At prepupa, the last larval stage, the larvae stop feeding in order to produce prothoracicotropic hormone (PPTH), which is necessary for metamorphosis. At this stage, they attain maximum size and have large protein and fat contents to sustain them through metamorphosis; they do not show significant changes in morphological characteristics at this stage [13]. Our results showed low consumption of feeds and weight gains of BSFL, indicating that the last four days can be considered as prepupal periods.

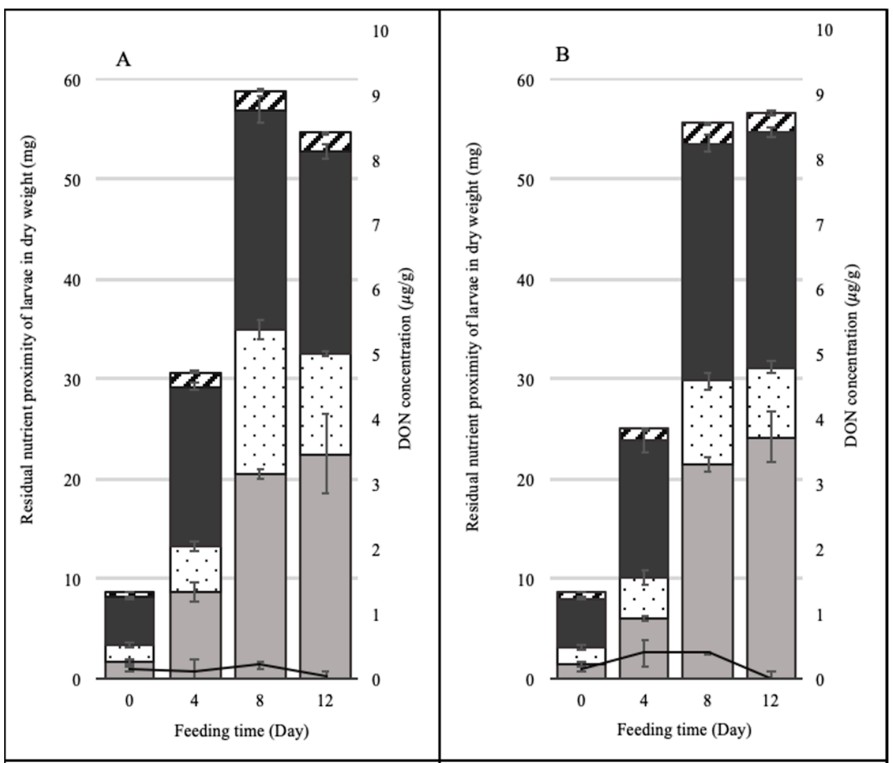

**Figure 2.** *Cont.*

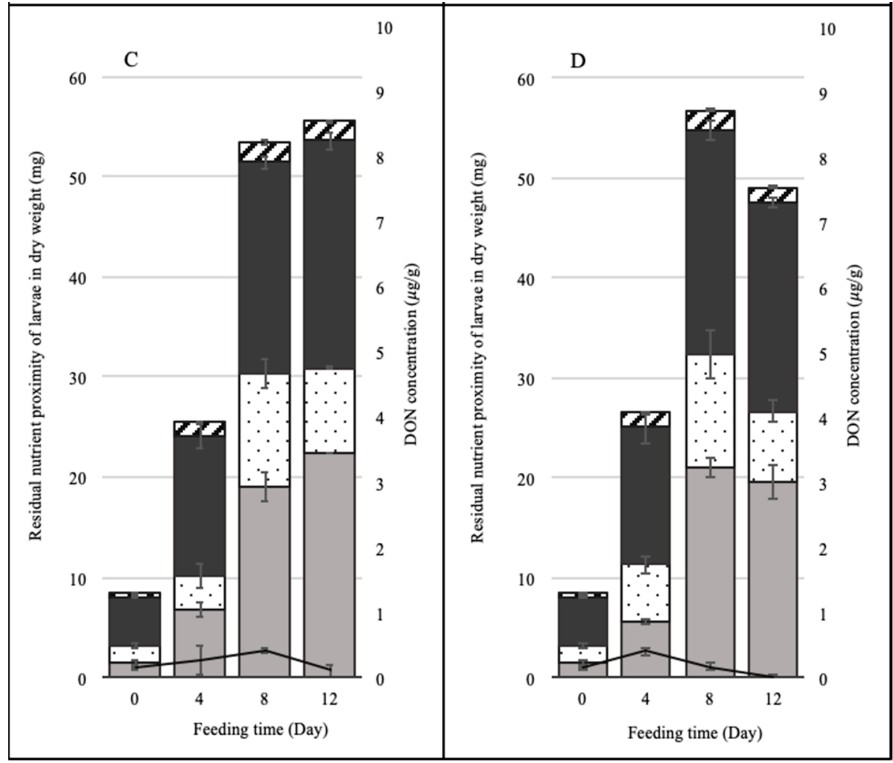

**Figure 2.** Weight gain, proximate composition, and deoxynivalenol (DON) concentration of larvae during black soldier fly larvae rearing. The overall length of each bar shows the dry weight of larvae divided according to the ratio of nutrients: hatched area, ashes; gray area, lipids; dotted area, carbohydrates; black area, proteins; the lines, DON concentration in black soldier fly larvae. Each panel represents the results of larvae from (**A**) Fusarium-damaged kernels (FDK) without inoculum (i.e., unfermented FDK), (**B**) FDK with *Lactobacillus plantarum*, (**C**) fermented FDK with *Aspergillus oryzae*, and (**D**), fermented FDK with both *L. plantarum* and *A. oryzae*.

### 3.2. The Proximate Composition of BSFL and Spent Feed

In order to achieve high larval growth, FDK was fermented for four days using generally regarded as safe (GRAS) strains (*Ao* and/or *Lp*). The nutrient profile differences among SSF treatments at day zero indicated that unfermented FDK contained 1.5% lipid, 8.9% protein, 2.2% ash, and 87.7% carbohydrates (Figure 1). At the fourth day, SSF did not change ash and lipid contents; however, the protein and carbohydrate contents differed. The protein content of the grain increased for all fermented FDK treatments; however, the carbohydrate content decreased during SSF for all treatments. While the carbohydrate ratio decreased, its absolute amount remained the greatest in the SSF feed, and the increased protein content was speculated to benefit larval growth. There was no difference in the amount of nutrients among microorganisms tested in this study; however, based on visual observation, FDK became softer compared to control FDK for easier mastication by BSFL.

During BSFL feeding trials, nutrient compositions of spent feed changed (Figure 1). Results showed that BSFL were able to consume and digest carbohydrates (i.e., starch) as their main food sources. The BSFL consumed around 23–25% of the dry matter in SSF feeds in 12 days, i.e., the residual amounts were reduced from 34 g at day zero to 25–26 g at day 12. The protein in spent feed decreased from $4.5 \pm 0.5$ to $1.8 \pm 0.2$ g on a dwb of FDK during BSFL digestion after 12 days of digestion for all SSF treatments. Lipid content of spent feed decreased significantly for all treatments after four days of digestion, and the decrease did not change by day 12. The carbohydrate content (i.e., starch) in spent feed decreased significantly during BSFL digestion for all fermented FDK. The ash content of spent feed decreased until the 12th day of digestion for the control, *Ao*, and *Lp* + *Ao* treatments. There was

no difference in the ash content of FDK fermented with *Lp*. The decrease in proximate composition of spent feed during BSFL digestion was associated with the consumption of nutrients by BSFL.

The proximate composition of larval biomass was determined (Figure 2). During feeding time, the ash amount of BSFL increased until day eight, when it reached the maximum level for all treatments, and then it remained constant until the 12th day. The initial ash content of BSFL was $0.5 \pm 0.0$ mg/larva and it reached $2.0 \pm 0.0$ mg/larva at day eight for FDK fermented with *Lp + Ao*. The carbohydrate content of BSFL was $1.7 \pm 0.2$ mg/larva, and, after the eight-day feeding period, the carbohydrate content increased to $14.6 \pm 1.0$, $8.3 \pm 0.8$, $11.3 \pm 1.5$, and $12.4 \pm 1.2$ mg/larva fed with FDK fermented in the control, *Lp, Ao*, and *Lp + Ao* treatments, respectively. The higher carbohydrate contents can be explained by an increase in the volume of BSFL, associated with an increase in larval skin chitin, which is produced during the growing stage of the BSFL. Kaya et al. [14] reported that chitin content increased gradually from larva to adult, and the highest chitin content was observed in adults. The carbohydrate content was higher in the control BSFL than among other treatments ($p < 0.05$). This can be explained by the reduced gain in the protein, lipid, and ash contents, i.e., concentrating of carbohydrates [10]. These results indicated that BSFL can recover the nutrients in damaged FDK and are not affected by the presence of mycotoxins.

Also, it was noticed that the sums of protein and lipid gains were $42.6 \pm 4.7$, $47.7 \pm 3.1$, $45.2 \pm 0.8$, and $40.4 \pm 2.1$ mg/larva fed with FDK fermented in the control, *Lp, Ao*, and *Lp + Ao* treatments, respectively, at day 12. At day eight, these figures were $42.3 \pm 1.8$, $45.2 \pm 1.6$, $40.1 \pm 2.1$, and $43.5 \pm 1.3$, respectively. Meanwhile the accumulation of carbohydrate at day eight (and day 12) were $14.6 \pm 1.0$ ($10.1 \pm 0.3$), $8.3 \pm 0.8$ ($7.0 \pm 0.6$), $11.3 \pm 1.5$ ($8.5 \pm 0.1$), and $11.3 \pm 2.3$, respectively. These results indicated that SSF grain-fed insects can achieve better protein and lipid profiles compared to unfermented grain-fed insects. It is, therefore, indicated that the SSF assisted to improve the nutrient components of the insect, while overall gain in weight in this study did not significantly vary among the feeds.

### 3.3. Amounts of DON in Spent Feed and BSFL Biomass

During BSFL digestion, the DON concentration in spent feed at the fourth day was the lowest compared to other days (Table 1). After four days of digestion, the concentration of DON in spent feed continuously increased until day 12. This increase in DON contents of the spent feed was comparable to the 23–25% consumption rate of nutrients by larvae. This suggested that the larvae did not assimilate DON in their body, but it was simply passed through their intestinal system. The highest DON concentration in the spent feed was for FDK fermented with *Lp*.

**Table 1.** Deoxynivalenol (DON) concentration in spent feed after black soldier fly larvae digestion.

| | DON Concentration (μg/g Spent Feed) * | | | |
| --- | --- | --- | --- | --- |
| | **Day 0** | **Day 4** | **Day 8** | **Day 12** |
| Control | $0.63 \pm 0.20$ [c,C] | $0.15 \pm 0.05$ [c,A] | $4.17 \pm 0.83$ [b,B] | $6.72 \pm 0.70$ [a,C] |
| *Lactobacillus plantarum* | $2.60 \pm 0.07$ [b,B] | $0.20 \pm 0.07$ [c,A] | $13.66 \pm 0.37$ [a,A] | $14.17 \pm 1.55$ [a,A] |
| *Aspergillus oryzae* | $2.76 \pm 0.27$ [b,B] | $0.12 \pm 0.02$ [c,A] | $5.80 \pm 0.88$ [a,B] | $6.82 \pm 1.82$ [a,B,C] |
| *L. plantarum + A. oryzae* | $3.58 \pm 0.18$ [b,A] | $0.12 \pm 0.05$ [b,A] | $10.69 \pm 1.77$ [a,A] | $11.41 \pm 2.25$ [a,A,B] |

\* Each value is expressed as the mean ± SD ($n = 3$). Statistical symbols: Means of DON concentration for each fermentation type marked with different lowercase letters (a, b, c) within a row are significantly different between days, and those followed by capital letters (A, B, C) within a column represent significant differences in DON concentration compared with fermentation type on the same day ($p < 0.05$).

There are several possibilities to explain the DON increase. Some microbial enzyme yielded during fermentation could enhance the DON levels, owing to a release of DON from kernel cell walls or other cell components [15]. Secondly, the DON may be produced during the feeding periods. Microorganism type and the proportion of each may result in an increase in mycotoxin production among microorganisms [5]. For example, the increase could also result from the increasing number of *Fusarium* spp. during fermentation time. Goral et al. [16] investigated the relationship between

concentration of *Fusarium* biomass and trichothecenes B (DON and nivalenol). They reported a stronger relationship between *Fusarium* biomass and DON content, as confirmed by FHB index. As another possibility, they might be converted from precursors. Nakagawa et al. [17] reported that deoxynivalenol-3-*O*-glucoside (D3G) has lower toxicity compared to its precursor DON, and D3G can be converted to DON in human or animal gut. Lactic acid bacteria have the capability to hydrolase the derivative D3G back to the toxic DON form. When the lactic acid bacteria ferment the feed, it might be responsible for the chemical conversion of conjugated mycotoxins and explain the high amount of DON in the spent feed [18]. While these factors can affect the results, it is unlikely appropriate to explain our results. We checked the DON amounts at each sampling time for SSF and unfermented grains. Such generations and conversions were counted in the data, regardless of their effect. We did not observe significant differences in DON amounts among the four conditions. Thus, we concluded that the differences in DON amounts between SSF and unfermented FDK were small and they did not significantly affect the contents of DON.

While DON amounts in the spent feed increased during the feeding period, BSFL biomass showed different trends in terms of DON amounts. In the growing stage, the larval biomass contained considerable DON concentration; however, there was no DON accumulation observed in BSFL after the 12-day rearing. Bosch et al. [3] evaluated the tolerance and accumulation of aflatoxin B1 (AFB1) in BSFL fed with AFB1-containing feeds to utilize the mycotoxin-contaminated crops. The BSFL did not contain detectable levels of AFB1 (<0.10 μg/kg). They suggested that the larvae rapidly excreted or metabolized the AFB1 after ingestion. Another possible reason was that part of the AFB1 could be found in bound form with proteins and left undetected. Camenzuli et al. [19] also investigated the potential of accumulation of AFB1, DON, ochratoxin A, and a mixture of mycotoxins in BSFL. None of the mycotoxins accumulated in the larval body; they were shown to excrete or metabolize the four mycotoxins present in the feed. Our results showed that BSFL growing on FDK feeds do not accumulate DON in their body. As shown in Figure 1, the larvae consumed most of the feed during the first eight days, and then consumption stopped by day 12, due to pupation. This is because they defecate their intestinal system and have an empty gut before pupation or shortly after adult emergence. The DON analysis indicated that BSFL do not assimilate DON in their body, and DON observed during the growth stage is speculated to be in the contents of their intestinal system.

## 4. Conclusions

The FDK used in this study had a large amount of nutrients, and BSFL converted these nutrients to insect biomass without accumulating DON in their bodies. While they consumed DON-contaminated materials, they contained DON in their body; however, the toxins were excreted from their body before they became pupae. They consumed ~2.8 g of FDK per g of BSFL body mass gained, and mainly converted starch into their proteins and lipids at a high efficiency. Thus, the BSFL can be used to separate nutrients from DON in FDK. Proximate analysis of fermented FDK showed higher protein and lipid contents, while there was no significant difference in the BSFL body mass gain among treatments. The procedure can be expanded to other mycotoxin-contaminated materials to recover valuable nutrients wasted in those contaminated materials. This waste treatment technology using BSFL may contribute to reducing the burden of animal protein shortages in the animal feed market and provide new income opportunities for small entrepreneurs in low- and middle-income countries.

**Author Contributions:** Conceptualization, T.T.; Methodology of fermentation, insect culture and proximate analyses, Z.G., S.A. and T.T.; Methodology of DON analysis, L.W. and H.R.K.; Investigation, Z.G., A.S., Y.B. and L.W.; Writing original draft preparation, Z.G.; Writing-Review & Editing, T.T.; Supervision, T.T.

**Funding:** This research received no external funding.

**Conflicts of Interest:** The authors declare no conflict of interest.

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
