# Peer review of "Deoxynivalenol (DON) Accumulation and Nutrient Recovery in Black Soldier Fly Larvae (Hermetia illucens) Fed Wheat Infected with Fusarium spp."

_fermentation, doi:10.3390/fermentation5030083_

Round 1

Reviewer 1 Report

The article is compact, but interesting, from the point of view of the possible using grain infected initially with mycotoxin such as DON, to obtain insect biomass without DON but with a balanced biochemical composition of proteins, carbohydrates and lipids, which can potentially be used as feed for poultry. It was shown that insects, assimilating the components of the grain, excrete mycotoxin. Thus, the plant (grain) biomass is transformed into animal biomass (insects) and freed from mycotoxin, since it is not absorbed into the digestive tract of the used insect. To better assimilate grain components and regulate the biochemical composition of insect biomass, the authors of the study conducted solid state fermentation (SSF) of contaminated kernels with the cells of filamentous fungi (Aspergillus oryzae) or/and lactic acid bacteria (Lactobacillus plantarum).

However, as follows from Figs. 1 and 2, the rate of grain consumption by insects, as well as the composition of insect biomass after 12 days and the absence of mycotoxin in this biomass by the end of the experiment are not strongly dependent on SSF in the presence of microorganisms. In the control (without any SSF) the result is not only similar, but even much better in comparison with the results of SSF conducted using lactic acid bacteria. I consider that this result is unpredictable and one of the main ones that should be reflected in Abstact and Conclusion.

According to the text itself, there are several recommendations addressed to the authors:

            - In the introduction, the initial motivation in favor of the need to conduct SSF should be strengthened, since it is very weak at the present time, and it is not clear from the introduction, whether and why it is necessary to conduct SSF and apply precisely those microorganisms that were specifically selected by the authors for this work. The use of such microorganisms usually supposes the biocatalytic degradation or sorption of DON. So, it is necessary to power the “motivation”;

            - in the Methods:

- The authors give information about "survival analysis (the number of larvae was counted at 0, 4, 8, and 12 days (initially exactly  50 larvae)... Larval volume was calculated by multiplying the length, width and thickness of 10 individual larvae before and after  digestion", but there are no data of such analysis in the text! Plese, give some explanation on a such information;

- the concentration of mycotoxin contaminating the grain (mg/ kg) initially should be clearly indicated. Now it’s not in the article, instead of it the following text is given: “The FDKs were soaked into water for 18 h at 21° C temperature. The soaked kernels were shred using a household coffee mill."

            - the correct cell concentration of both cultures used in SSF should be clearly indicated. That will be correct from microbiological point of view. Now it is not in the article,

            In results:

- it is necessary to give a control of the rate of grain consumption in the absence of mycotoxin contamination, now there is no such control. It is necessary to understand how does DON decrease this characteristic;

            - in the captions to figures 1 and 2, it is necessary to completely decipher what control is, Ao and Lp;

            - in Table 1, please, give a clearer explanation of what ABCabc is;

            - references 15 and 16 are quite “old” for the field of science in which the presented study was carried out, so it is better to replace them with more “fresh” ones. I suggest the authors themselves choose a replacement.

Finally, I have no idea how did the authors carefully separated Larval biomass from spent feed for the analysis of DON. There is no something concerning such "separation" in Methods.  

Author Response

Reviewer 1

Comments and Suggestions for Authors

The article is compact, but interesting, from the point of view of the possible using grain infected initially with mycotoxin such as DON, to obtain insect biomass without DON but with a balanced biochemical composition of proteins, carbohydrates and lipids, which can potentially be used as feed for poultry. It was shown that insects, assimilating the components of the grain, excrete mycotoxin. Thus, the plant (grain) biomass is transformed into animal biomass (insects) and freed from mycotoxin, since it is not absorbed into the digestive tract of the used insect. To better assimilate grain components and regulate the biochemical composition of insect biomass, the authors of the study conducted solid state fermentation (SSF) of contaminated kernels with the cells of filamentous fungi (Aspergillus oryzae) or/and lactic acid bacteria (Lactobacillus plantarum).

However, as follows from Figs. 1 and 2, the rate of grain consumption by insects, as well as the composition of insect biomass after 12 days and the absence of mycotoxin in this biomass by the end of the experiment are not strongly dependent on SSF in the presence of microorganisms. In the control (without any SSF) the result is not only similar, but even much better in comparison with the results of SSF conducted using lactic acid bacteria. I consider that this result is unpredictable and one of the main ones that should be reflected in Abstact and Conclusion.

Our response

Thank you for your comments. We realized the effects of SSF is not significant in this study. In our previous study, we showed the SSF helps to increase the insect biomass gains through increasing lipids and proteins in the feed. The feeds in these studies were low-starch materials in contrast to the present research, starchy cereal grains. Our observation reflected this difference. The starchy materials can support the growth of insects through high availability of carbohydrates. Their overall weight gain are similar among the feeds. But we like to emphasize the amount of proteins are higher in SSF-grain fed insect. We include this argument in line 23-25, 270-277 and 334-336.

According to the text itself, there are several recommendations addressed to the authors:

            - In the introduction, the initial motivation in favor of the need to conduct SSF should be strengthened, since it is very weak at the present time, and it is not clear from the introduction, whether and why it is necessary to conduct SSF and apply precisely those microorganisms that were specifically selected by the authors for this work. The use of such microorganisms usually supposes the biocatalytic degradation or sorption of DON. So, it is necessary to power the “motivation”;

Our response

We have added a paragraph to argue the background of SSF utilization in an added paragraph in Introduction (Line 70-77).

            - in the Methods:

The authors give information about "survival analysis (the number of larvae was counted at 0, 4, 8, and 12 days (initially exactly  50 larvae)... Larval volume was calculated by multiplying the length, width and thickness of 10 individual larvae before and after  digestion", but there are no data of such analysis in the text! Plese, give some explanation on a such information;

Our response

In line 212-214 (the second paragraph in Section 3.1.), we discuss the initial and final volume of BSFL.

- the concentration of mycotoxin contaminating the grain (mg/ kg) initially should be clearly indicated. Now it’s not in the article, instead of it the following text is given: “The FDKs were soaked into water for 18 h at 21° C temperature. The soaked kernels were shred using a household coffee mill."

Our response

The information about initial concentrations of DON in wheat grain is added in the material-method section (Line 100; the first line of Section 2.2.).

            - the correct cell concentration of both cultures used in SSF should be clearly indicated. That will be correct from microbiological point of view. Now it is not in the article,

Our response

We realized it, but the concentration of active fungi cells are difficult to determine. Usually we express the inoculation of fungi as spore/ml inoculation. While it assures the accuracy of inoculum amounts, it makes the delay of growth (for germination of spores). To maintain the experimental design comparable between bacterium (Lactobacillus plantarum) and fungus (Aspergillus oryzae), we intentionally plan the experiments to use the fully grown preculture, rather than define the cell numbers for both fungus and bacterium.

            In results:

- it is necessary to give a control of the rate of grain consumption in the absence of mycotoxin contamination, now there is no such control. It is necessary to understand how does DON decrease this characteristic;

Our response

We appreciate this comment. We admit the lack of such reference. However, the inclusion of a reference in that regard raises another question: How to standardize between  FDK-grains and undamaged grains. They would have different amount of carbohydrates, proteins, lipids, and ashes, and different physical conditions. Thus we think the inclusion of that complexes the arguments.

            - in the captions to figures 1 and 2, it is necessary to completely decipher what control is, Ao and Lp;

Our response

We revised the figure legends accordingly. The legends now indicate the panel with proper designations.

            - in Table 1, please, give a clearer explanation of what ABCabc is;

Our response

The footnote of Table 1 includes the explanation of these statistical letters.

            - references 15 and 16 are quite “old” for the field of science in which the presented study was carried out, so it is better to replace them with more “fresh” ones. I suggest the authors themselves choose a replacement.

Our response

Reference 15 and 16 were substituted with more recent studies.

Finally, I have no idea how did the authors carefully separated Larval biomass from spent feed for the analysis of DON. There is no something concerning such "separation" in Methods.  

Our response

We now include “BSFL Separation After Digestion” subtitle in Material-Method section. Briefly, the larvae were separated from the spent feeds using forceps, and the residual feeds on the surface of larvae were rinsed off. (Please note that the washed-off feeds were put back to the spent feed to minimize the errors in spent feed data.) The contents of larval digestive tracts cannot be separated without affecting the insect growth, thus the data of insects include the contents of GI tract. We argue this inclusion is the actual detection of DON during the growth of BSFL.

Submission Date

22 August 2019

Date of this review

02 Sep 2019 15:57:17

Reviewer 2 Report

Deoxynivalenol (DON) accumulation and nutrient 1 recovery in black soldier fly larvae (Hermetia  illucens) fed wheat infected with Fusarium spp.

The topic is very interesting and within the scope of this Journal.

It is just for information, aflatoxins and cyclopiazonic acid were also cheked in these samples. These mycotoxins are also produced by A. oryzae and I wanted to know if you have checked these mycotoxin levels as well, it would be interesting to include these results in this article.

Recent authors have studied aflatoxins in Hermetia  illucens, I think this reference : “Aflatoxin B1 tolerance and accumulation in Black Soldier Fly Larvae (Hermetia illucens) and Yellow

Mealworms (Tenebrio molitor)” may be included in this article and also this one related to DON: “Tolerance and excretion of the mycotoxins Aflatoxin B1, Zearalenone, Deoxynivalenol, and Ochratoxin A by Alphitobius diaperinus and Hermetia illucens from contaminated substrates”.

Abstract

Line 15: Fusarium can be written in italics

Introduction

Line 39: Please specify 1 ppm in measurement units

Line 49: “Alternate approaches should be considered to utilize inedible FHB-damaged kernels (FDK).” Which is your proposal??I think your proposal, suggestion, may be considered here.

Lines between 50 and 54:  “World population …environmental damage” I think that do not contribute to additional information and also reference is not provided.

There are no references in relation to specific insect regulation along the introduction and what is the destination, please, specify. I would rather point that there is no legislation in relation to mycotoxins and this type of foodstuffs.

Material and methods

Line 89 and 96: Instrument references are not homogenous…please be concise. In the first one, i.e. : “Woodbridge, ON, Canada” and in the second one city and country were not included…: “Sorvall GSA rotor”

Line 130: “Goldfisch apparatus” intruments references are not completed…please check it along the text…Solvents were neither well referenced…

Results and discussion

Very interesting results.

Line 252: “Lactic acid bacteria have the capability to hydrolase (is not hydrolyze in English?) the derivative deoxynivalenol-3-β-d-glucoside back to the toxic DON form”. Please if you have results related to these affirmation include it into the text. It is very important. Did you check DON derivatives in these samples?

Line 266 and 267: The DON concentration in BSFL biomass was determined after feeding on fermented FDK (Figure 2). This sentence is incorrect here because you talked about it before and also this figure was developed in previous sentences

I miss some articles to compare DON accumulation in insects in the discussion…be more concise in the discussion, you have to look up more references to compare your results well.

References

The numerical order is repeated in all references.

References are not carefully reviewed by the authors.

I think these references are not enough, please include more…i.e. as I suggested you at the beginning of the review.

Line 290: The correct abbreviation is: J. Anim. Physiol. Anim. Nutr.The abbreviation iIs not well written : “J. Anim. Physiol. An. N. “

Line 294: Fusarium is written in italics and capital letter: “fusarium head blight “

Line 295: The Journal is wrong….please check it again…

Line 297: Journal abbreviation is not ok, te correct acronym is : J. Waste Manag. Please review all references.

Line 314: Authors are not complete. “Adam G.” is not reported and volumen is also missing.

Author Response

Reviewer 2:

Comments and Suggestions for Authors

Deoxynivalenol (DON) accumulation and nutrient recovery in black soldier fly larvae (Hermetia  illucens) fed wheat infected with Fusarium spp.

The topic is very interesting and within the scope of this Journal.

It is just for information, aflatoxins and cyclopiazonic acid were also cheked in these samples. These mycotoxins are also produced by A. oryzae and I wanted to know if you have checked these mycotoxin levels as well, it would be interesting to include these results in this article.

Recent authors have studied aflatoxins in Hermetia  illucens, I think this reference : “Aflatoxin B1 tolerance and accumulation in Black Soldier Fly Larvae (Hermetia illucens) and Yellow Mealworms (Tenebrio molitor)” may be included in this article and also this one related to DON: “Tolerance and excretion of the mycotoxins Aflatoxin B1, Zearalenone, Deoxynivalenol, and Ochratoxin A by Alphitobius diaperinus and Hermetia illucens from contaminated substrates”.

Our response:

 These two references were added through the text.

Abstract

Line 15: Fusarium can be written in italics

Our response:

We corrected it.

Introduction

Line 39: Please specify 1 ppm in measurement units

Our response:

We corrected it.

Line 49: “Alternate approaches should be considered to utilize inedible FHB-damaged kernels (FDK).” Which is your proposal??I think your proposal, suggestion, may be considered here.

Our response:

We added a statement about our alternate approach studied in this manuscript (line 51-52).

Lines between 50 and 54:  “World population …environmental damage” I think that do not contribute to additional information and also reference is not provided.

Our response:

We added the reference #4.

There are no references in relation to specific insect regulation along the introduction and what is the destination, please, specify. I would rather point that there is no legislation in relation to mycotoxins and this type of foodstuffs.

Our response:

The regulations vary among countries and areas, but increasing awareness of entomophagy gives the rationale to study the utilization of waste materials converting it to edible proteins and lipids. We added a reference (#6) about entomophagy to support our argument in line 63-65.

Material and methods

Line 89 and 96: Instrument references are not homogenous…please be concise. In the first one, i.e. : “Woodbridge, ON, Canada” and in the second one city and country were not included…: “Sorvall GSA rotor”

Our response:

We corrected them.

Line 130: “Goldfisch apparatus” intruments references are not completed…please check it along the text…Solvents were neither well referenced…

 Our response:

The supplier information of goldfisch apparatus was added.

Results and discussion

Very interesting results.

Line 252: “Lactic acid bacteria have the capability to hydrolase (is not hydrolyze in English?) the derivative deoxynivalenol-3-β-d-glucoside back to the toxic DON form”. Please if you have results related to these affirmation include it into the text. It is very important. Did you check DON derivatives in these samples?

Our response:

We checked the DON amounts at each sampling time for SSF- and unfermented-grains. Thus the conversion of " the derivative deoxynivalenol-3-β-d-glucoside back to the toxic DON form" was counted in the data, even if it affects. The data indicated that the differences between SSF- and unfermented-grains are small and not significantly affect the contents of DON. We tried to discuss all possibilities which may be resulted in higher DON concentration in spent feed after fermentation (Line 294-312).

Line 266 and 267: The DON concentration in BSFL biomass was determined after feeding on fermented FDK (Figure 2). This sentence is incorrect here because you talked about it before and also this figure was developed in previous sentences

Our response:

We have deleted the statement.

I miss some articles to compare DON accumulation in insects in the discussion…be more concise in the discussion, you have to look up more references to compare your results well.

Our response:

Four references were added in Discussion section (Line 294-312; 316-323).

References

The numerical order is repeated in all references.

Our response:

We checked the list.

References are not carefully reviewed by the authors.

Our response:

We checked all references to make sure the correct references are cited.

I think these references are not enough, please include more…i.e. as I suggested you at the beginning of the review.

Our response:

Three references were added through the text. We think they are sufficiently support our arguments.

Line 290: The correct abbreviation is: J. Anim. Physiol. Anim. Nutr.The abbreviation iIs not well written : “J. Anim. Physiol. An. N. “

Our response:

We corrected it.

Line 294: Fusarium is written in italics and capital letter: “fusarium head blight “

Our response:

We corrected it.

Line 295: The Journal is wrong….please check it again…

Our response:

We corrected it.

Line 297: Journal abbreviation is not ok, te correct acronym is : J. Waste Manag. Please review all references.

Our response:

We corrected it.

Line 314: Authors are not complete. “Adam G.” is not reported and volumen is also missing.

Our response:

We corrected it.

Submission Date

22 August 2019

Date of this review

31 Aug 2019 23:05:54

Round 2

Reviewer 1 Report

The authors have made a notable work with the text. I am satisfied with all given explanations and consider that the second version of the manuscript became better than previous one.